# Analysis of the Volatile Flavor Compounds of Pomegranate Seeds at Different Processing Temperatures by GC-IMS

**DOI:** 10.3390/molecules28062717

**Published:** 2023-03-17

**Authors:** Lili Gao, Lihua Zhang, Jing Liu, Xiao Zhang, Yonghui Lu

**Affiliations:** 1School of Pharmacy, Shandong University of Traditional Chinese Medicine, Jinan 370113, China; 2College of Food Science and Pharmaceutical Engineering, Zaozhuang University, Zaozhuang 370402, China

**Keywords:** pomegranate seeds, processing temperature, volatile components, GC-IMS

## Abstract

This study sought to reveal the mechanism of flavor generation when pomegranate seeds are processed, as well as the contribution of volatile organic components (VOCs) to flavor formation. Gas chromatography–ion mobility spectrometry (GC-IMS), combined with relative odor activity (ROAV) and statistical methods, was used for the analysis. The results showed that 54 compounds were identified from 70 peaks that appeared in the GC-IMS spectrum. Then, the ROAV results showed 17 key volatile components in processing pomegranate seeds, and 7 flavor components with large differential contributions were screened out using statistical methods. These included γ-butyrolactone, (E)-3-penten-2-one (dimer), pentanal, 1-propanethiol, octanal, and ethyl valerate (monomer). It is suggested that lipid oxidation and the Maillard reaction may be the main mechanisms of flavor formation during the processing of pomegranate seeds. Furthermore, this study lays the experimental and theoretical foundations for further research on the development of flavor products from pomegranate seeds.

## 1. Introduction

Pomegranates (*Punica granatum* L.) belong to the family Punicaceae, which is a deciduous shrub or small tree. It originated from the Himalayas and Mediterranean region and grows mainly in tropical and subtropical countries such as India, Iran, China, and the USA [1,2]. The pomegranate is not only of high economic, nutritional, medicinal, and ornamental value, but also plays an important role in the industrial, cultural, and ecological fields. Some research has revealed that pomegranate is utilized worldwide because its different parts (fruit juice, peel and seeds, flowers, leaves, bark) are rich in bioactive constituents [3], and related industries are expanding rapidly. Pomegranate seeds are a by-product of pomegranate juice, accounting for about 20% of the total fruit weight, and they play an important role as antioxidants, anti-free radicals, and adjusters of immunity, as well as having anti-tumor, anti-osteoporosis, anti-cardiovascular disease effects; they also have capabilities related to unsaturated fatty acids, phenols, sterols, proteins, and volatile oils, among others [4,5,6,7]. Similarly, pomegranate seeds are an interesting source of a rare conjugated linoleic acid, which may be useful in the prevention and treatment of many diseases, especially diabetes, arteriosclerosis, obesity, and cancer (breast, prostate, and colon) [8]. Notably, during the experiment, we found that pomegranate seeds gave off different levels of odor at different roasting temperatures. Gas chromatography—ion mobility spectrometry (GC-IMS) is a new technology that is widely used to analysis volatile organic components (VOCs). GC-IMS combines the high separation ability of GC and the rapid response of IMS, and has high sensitivity, powerful functions, high accuracy, a low detection limit, and no requirement for sample pretreatment, and is simple to operate. Above all, the original flavor of the sample is retained to the maximum extent [9].

Currently, the research on pomegranate seeds focuses on the extraction process, the identification and separation of chemical components, pharmacological effects, and biological activities. The blanching and microwave preprocessing of pomegranate seeds prior to pressing has been reported to enhance oil yield, total phenolic content, flavor compounds, and DPPH and ABTS radical scavenging capacity [4]. Additionally, the roasting technique can also cause significant fluctuations in phenolic and fatty acid content [10]. Zhang et al. [11] discovered that pomegranate seeds emit a strong sesame fragrance when baked at high temperatures. Min et al. [6] identified volatile oil components in pomegranate seeds using GC-IMS. Nonetheless, there are few reports on the mechanism of flavor formation and the VOCs of pomegranate seeds according to processing at different temperatures. In this paper, the contribution of different volatile flavor components of pomegranate seeds at various processing temperatures was analyzed and confirmed using GC-IMS technology combined with relative odor activity (ROAV), principal component analysis (PCA), and orthogonal partial least squares discriminant analysis (OPLS-DA). Furthermore, this study sought to reveal the mechanism of flavor formation of processed pomegranate seeds.

## 2. Results and Analysis

### 2.1. Topographic Plot of Pomegranate Seeds at Different Processing Temperatures

GS-IMS was used to analyze the differences in volatile flavor components of pomegranate seed samples at various processing temperatures. GC-IMS analysis of the volatile components resulted in a 3D topographic plot using the Reporter plug-in, where the X, Y, and Z-axes represent the drift time (DT) of the drift tube, the residence time (RT) in the capillary gas chromatography column, and the ion peak intensity for quantification.

As shown in Figure 1A, the peak signal distribution of samples was basically the same, but the intensity was different. This indicated that the VOCs of pomegranate seed samples were very similar at different temperatures, but the VOC content was different. At the same time, the differential comparison mode was used to show the differences between the samples. As shown in Figure 1B, the topographic plot of the CK (raw pomegranate seeds) group was selected as a reference. If the VOCs were consistent, the background after deduction was white, while red indicated that the concentration of the substance was higher than that of the reference, and blue indicated that the concentration of the substance was lower than that of the reference. It can be seen that the VOCs of pomegranate seed samples have different spectral information at different temperatures. The results demonstrated that the difference between group A (pomegranate seeds processed at 140 °C) and CK was the smallest, while the difference between groups B (pomegranate seeds processed at 160 °C), C (pomegranate seeds processed at 180 °C), and D (pomegranate seeds processed at 200 °C) and CK was significant, but the difference between group D was the largest. Additionally, there is no obvious difference between B and C.

### 2.2. Fingerprints of Volatile Compounds in Pomegranate Seeds at Different Processing Temperatures

In order to correctly assess the closely connected substances on the topographic plot, all the information given by the fingerprint analysis technique was used for qualitative characterization. In the fingerprint (Figure 2), each row shows all the signal peaks of a sample, and each column shows the same volatile compound in different samples. Additionally, its color represents the content of volatile compounds, with a brighter color indicating higher content. As shown in Figure 2, the contents of volatile organic compounds were very different in the samples of each group. A total of 70 VOCs were displayed, of which 54 VOCs were clearly identified; these are named after compounds, and the unidentified VOCs are represented by the numbers 1–16. Identified compounds were divided into five categories, comprising 17 aldehydes, 16 esters, 8 ketones, 7 alcohols, and 6 other compounds, shown in Table 1. Fourteen compounds produced two signals with different intensities due to differences in concentration or proton affinity, including benzaldehyde, (E,E)-2,4-hexadienal, hexanal, furfural, butyl butyrate, butyl propionate, ethyl valerate, 2-methylbutanol acetate, 2-methylbutyl formate, ethyl propionate, 2-hexanone, (E)-3-pentene-2-one, and 1-hexanol.

As shown in Figure 2, there are obvious differences in the VOCs of pomegranate seeds with processing temperatures. The greatest difference was observed between groups D and CK, with D having the most bright red spots, and CK having the fewest. Several representative regions were divided according to the differences in VOC content. Region I was a similar region of VOCs, including hexanal (monomer), 2-methylbutyl acetate (monomer and dimer), 2-methylbutyl formate (monomer and dimer), ethyl valerate (monomer and dimer), and butyl propionate (monomer and dimer); II was the characteristic region of group CK, including hexanal (dimer), ethyl acetate, ethyl propionate (monomer and dimer), 1-hexanol (monomer and dimer) and nonanal. Region III was the characteristic region of group A, including 2-methyl-1-butanol, 3-methyl-1-butanol, 1-pentanol, methacrolein, and 1-propanethiol, while IV was the characteristic region of group D, which not only included furfural (monomer and dimer), 2,5-dimethylpyrazine, trimethylpyrazine, pyrrole, γ-butyrolactone, and heterocyclic compounds, but also included ethyl acetate, 2-butanone, (E,E)-2, 4-hexadienal, 1-propanol and dimethyl disulfide. Nevertheless, there were no characteristic regions in groups B and C because of no significant differences in content.

As shown in Table 1, the area normalization method was applied to calculate the relative content of each component with the peak volume of VOCs, and a clustering analysis was conducted. High contents of acetone, 2-methylbutyl formate (dimer), and ethyl valerate (dimer) were detected in all samples, which may contribute little to distinguishing the different aromas of processed pomegranate seeds. There were relatively high contents of ethyl acetate and hexanal (dimer) in group CK, ethyl valerate (monomer) and butyl propionate (dimer) in group A, and butyraldehyde and 2-methyl-1-butanal in groups B, C, and D. These may be the compounds that contribute to the aroma of each group. Moreover, with the rise in temperature, the esters showed a downward tendency, while the aldehydes showed an upward to downward tendency. Likewise, the change tendency of alcohols and other components is consistent with that of ketones, being inclined from upward to downward to upward, but this tendency is not significant, especially for alcohols. For details, see Figure 3.

### 2.3. Analysis of the Flavor Contribution of Pomegranate Seed VOCs

ROAV [12] is used to estimate the contribution of flavor compounds to the overall odor and to determine the most effective odor. Table 2 shows the results of ROAV. There were 11, 13, 15, 17, and 15 key flavor compounds and there were 11, 10, 7, 6, and 8 embellished flavor compounds in groups CK, A, B, C, and D, respectively. Moreover, the ROAV values of nonanal, octanal, hexanal (dimer), 2-methyl-1-butanal, 3-methyl-1-butanal, ethyl valerate (monomer and dimer), 2-methybutyl acetate (monomer and dimer), and ethyl acetate were greater than 1 in all groups. The ROAV values of (E)-2-heptenal, butyl propionate (monomer and dimer), (E)-3-penten-2-one (monomer), and 2-butylfuran were between 0.1 and 1 in all groups. Therefore, there was little difference in the contribution of the above compounds to the flavor. However, hexanal (monomer) of group CK, hexanal (monomer) and 1-propanethiol of group A, hexanal (monomer), pentanal, butyraldehyde, and γ-butyralactone of group B, hexanal (monomer), pentanal, butanaldehyde, γ-butyralactone, (E)-3-pentene-2-one (dimer), and dimethyl disulfide of group C, pentanal, butanaldehyde, γ-butyralactone, (E)-3-pentene-2-one (dimer), and dimethyl disulfide of group D were also greater than 1, and they played a key role in the formation of the pomegranate seed flavor in each group. Moreover, for 2-hexenal, pentanal, butyl butyrate, γ-butyralactone, and (E)-3-pentene-2-one (dimer) in group CK, 2-hexenal,pentanal, n-butanal, γ-butyralactone, and (E)-3-pentene-2-one (dimer) in group A, 2-hexenal and (E)-3-pentene-2-one (dimer) in group B, 2-hexenal in group C, and n-hexanal, trimethylpyrazine, and pyrrole in group D, the ROAV values were also in the range of 0.1~1, and they played an auxiliary role in the formation of the pomegranate seed flavor in each group.

**Table 2 molecules-28-02717-t002:** Aroma threshold, aroma characteristics, and relative odor activity (ROAV) in pomegranate seeds.

No.	Compound	Odor Thresholda (μg/kg)	Aroma Description	ROVA					References
				CK	A	B	C	D	
1	nonanal	1	citrus-fishy, waxy, fresh, fatty, orris, grapefruit, aldehydic, orange peel, lime, rose, green	6.5109	3.4281	3.2480	3.0786	2.1879	[13,14]
2	octanal	0.7	lemon, citrus, soap, orange peel, waxy, fatty, aldehydic, green	2.3550	1.7161	2.3773	2.0247	1.4756	[13,14]
3	benzaldehyde	350	almond, burnt sugar	0.0189	0.0196	0.0139	0.0154	0.0114	[13,15]
4	benzaldehyde *	350	almond, burnt sugar	0.0025	0.0030	0.0018	0.0025	0.0017	[13,15]
5	(E)-2-heptenal	13	soap, fat	0.1245	0.2756	0.2163	0.1923	0.1944	[13]
6	(E,E)-2,4-hexadienal	94.8	green, floral	0.0032	0.0038	0.0064	0.0141	0.0131	[13]
7	(E,E)-2,4-hexadienal *	94.8	green, floral	0.0010	0.0007	0.0007	0.0008	0.0033	[13]
8	hexanal	4.5	leafy, grass, sweaty, tallow, fresh, fatty, fruity, aldehydic, green	1.8454	1.7687	1.3454	1.0901	0.8104	[13]
9	hexanal *	4.5	leafy, grass, sweaty, tallow, fresh, fatty, fruity, aldehydic, green	6.5146	1.5343	2.3180	1.8512	1.1149	[13,14]
10	(E)-2-hexenal	17	leafy, apple, cheesy, vegetable, banana, rancid, fatty, sweet, plum, fruity, aldehydic, almond, green	0.1491	0.1000	0.1186	0.1127	0.0549	[14]
11	furfural	282	fragrant, bread, woody, sweet, baked, almond	0.0020	0.0032	0.0073	0.0108	0.0203	[13,14]
12	furfural *	282	fragrant, bread, woody, sweet, baked, almond	0.0077	0.0074	0.0057	0.0091	0.0357	[13,14]
13	pentanal	12	bready, fermented, berry, malt, pungent, fruity, nutty, almond	0.8320	0.7529	1.7159	1.3498	1.1757	[13,14]
14	2-methyl-1-butanal	1	musty, coffee, cocoa, nutty, almond, fermented	3.9212	14.5229	26.6329	24.2428	21.4276	[13,14]
15	3-methyl-1-butanal	0.2	peach, sour, chocolate, ethereal, malt, fatty, aldehydic, rancid, pungent	4.7358	37.3441	100.0000	92.6050	78.2685	[13,14]
16	butyraldehyde	9	cocoa, green, fermented, pungent, green	0.0731	0.5921	3.4533	3.9286	3.0752	[14]
17	methacrolein	NA	pungent	NA	NA	NA	NA	NA	NA
	aldehydes	NA	NA	27.0969	62.0726	141.4615	130.5283	109.8707	NA
18	butyl butyrate	400	fruity, banana, pineapple	0.0121	0.0153	0.0128	0.0139	0.0124	[16]
19	butyl butyrate *	400	fruity, banana, pineapple	0.0014	0.0021	0.0019	0.0024	0.0024	[16]
20	butyl propionate	25	fruity, apple-like	0.4104	0.4758	0.3801	0.3179	0.2570	[17]
21	butyl propionate *	25	fruity, apple-like	0.4709	0.6430	0.4635	0.4385	0.3778	[17]
22	ethyl valerate	1.5	fruity, sweet, pineapple	12.2372	12.2845	9.2911	8.3215	6.6979	[18]
23	ethyl valerate *	1.5	fruity, sweet, pineapple	24.1929	27.4440	19.5073	18.5524	15.5167	[18]
24	2-methybutyl acetate	5	apple-like	2.8490	2.1130	1.7931	1.4343	1.1536	[19]
25	2-methybutyl acetate *	5	apple-like	2.9779	1.7807	1.6542	1.2712	1.4058	NA
26	methyl 3-methylpentanoate	NA	NA	NA	NA	NA	NA	NA	NA
27	2-methylbutyl formater	NA	NA	NA	NA	NA	NA	NA	NA
28	2-methylbutyl formate *	NA	NA	NA	NA	NA	NA	NA	NA
29	ethyl propionate	10	fruity, pineapple	0.3879	0.0817	0.0615	0.0778	0.0824	[18]
30	ethyl propionate *	10	fruity, pineapple	0.6328	0.0357	0.0661	0.0500	0.0438	[18]
31	ethyl acetate	5	fruity sweet	7.2821	2.6585	1.1707	1.0254	1.0882	[19]
32	γ-butyrolactone	1.1	sweet, buttery	0.9437	0.9597	1.0565	1.7893	4.7373	[20]
33	methyl acetate	470	fruity	0.0042	0.0109	0.0122	0.0149	0.0231	[21]
	esters	NA	NA	43.1517	44.7583	33.1040	30.3522	25.4237	NA
34	6-methyl-5-hepten-2-one	50	pepper, apple, mushroom, citrus, musty, rubber, nutty, green, hazelnut, bitter, lemongrass	0.0420	0.0368	0.0334	0.0302	0.0277	[13,14]
35	2-hexanone	NA	floral, apple-like	NA	NA	NA	NA	NA	[22]
36	2-hexanone *	NA	floral, apple-like	NA	NA	NA	NA	NA	[22]
37	cyclopentanone	NA	pleasing mint	NA	NA	NA	NA	NA	[23]
38	(E)-3-penten-2-one	1.5	acetone, fishy, fruity, phenolic	0.1571	0.1080	0.2853	0.5177	0.5784	[24]
39	(E)-3-penten-2-one *	1.5	acetone, fishy, fruity, phenolic	0.1293	0.2356	0.5314	1.2378	1.4008	[24]
40	2-butanone	50,000	ether, fragrant, fruit, pleasant, sweet	0.0000	0.0000	0.0001	0.0002	0.0004	[25]
41	acetone	500,000	minty chemical, sweet, solventy	0.0001	0.0001	0.0001	0.0001	0.0001	[26]
	ketones			0.3286	0.3805	0.8503	1.7860	2.0075	NA
42	1-hexanol	50	oil, alcoholic, ethereal, resinfusel, sweet, fruity, flower, green	0.0208	0.0112	0.0030	0.0018	0.0013	[13,14]
43	1-hexanol *	50	oil, alcoholic, ethereal, resin, fusel, sweet, fruity, flower, green	0.0417	0.0086	0.0039	0.0057	0.0083	[13,14]
44	1-pentanol	5000	oil, balsamic, vanilla, fusel, sweet, balsam	0.0009	0.0014	0.0010	0.0007	0.0005	[13,14]
45	1-pentanol *	5000	oil, balsamic, vanilla, fusel, sweet, balsam	0.0005	0.0011	0.0006	0.0005	0.0004	[13,14]
46	2-methyl-1-butanol	300	NA	0.0008	0.0020	0.0009	0.0005	0.0005	[19]
47	3-methyl-1-butanol	300	oil, alcoholic, burnt, whiskey, malt, banana, fusel, fruity	0.0013	0.0041	0.0017	0.0014	0.0013	[14,19]
48	1-propanol	9000	alcohol	0.0001	0.0006	0.0011	0.0013	0.0016	[18]
	alcohols	NA	NA	0.0660	0.0290	0.0122	0.0120	0.0140	NA
49	trimethylpyrazine	28	roasted, coffee, cocoa	0.0272	0.0277	0.0252	0.0447	0.1025	[13,27]
50	2,5-dimethylpyrazine	20	popcorn, roasted	0.0292	0.0371	0.0393	0.0516	0.0633	[13]
51	pyrrole	20	nut, sweet	0.0302	0.0304	0.0279	0.0718	0.2424	[25]
52	1-propanethiol	2	onion	0.2816	2.4936	0.5941	0.5386	0.5159	[26]
53	dimethyl disulfide	1.1	onion, cabbage, putrid	0.6252	0.4915	0.7347	2.8772	6.8356	[13,19]
54	2-butylfuran	5	mild, wine, sweet, fruity, spicy	0.1215	0.2212	0.2874	0.3330	0.3167	[14]
	others	NA	NA	1.1149	3.3015	1.7086	3.9168	8.0764	NA

* represents the dimer of the compound; NA indicates that it is not recorded in the literature.

Aldehydes with grass, nuts, candy, or dried cheese have a low threshold, which made a relatively large contribution to the flavor of the sample. There are mainly C_4_–C_9_ aldehydes in pomegranate seeds. With the increase in temperature, the ROAVs of pentanal, butyraldehyde 2-methyl-1-butanal, and 3-methyl-1-butanal with roasted or nutty flavors increased, while the ROAVs of nonanal, octanal, and hexanal with a fatty flavor decreased. Esters with fruity or floral aromas are widespread in fruits. Pomegranate seeds, as part of the pomegranate fruit, are rich in esters with high content. The ROAVs of most esters decreased to different degrees, except for methyl acetate and γ-butyralactone. Ketones with a distinctive fragrance and fruity aroma also contributed to the flavor. However, the ROAVs of (E)-3-pentene-2-one were increased, while those of melatonin decreased. Alcohol is widespread in nature with low relative content and a high threshold in pomegranate seeds, which has little effect on the formation of flavor. The aroma thresholds of sulfur compounds and nitrogen compounds were lower, which had a greater influence on the formation of the aroma. Sulfur-containing compounds are easy to volatilize, and pyrazines can provide different flavors depending on the type of alkyl substituent [28]. The ROAVs of 1-propanethiol, dimethyl disulfide, 2-butylfuran, pyrrole, 2,5-dimethylpyrazine, and trimethylpyrazine in pomegranate seeds increased with temperature. In summary, the processing of pomegranate seeds can contribute to the generation of flavor.

### 2.4. Principal Component Analysis (PCA) and Orthogonal Partial Least Squares Discriiminate Analysis (OPLS-DA)

In order to differentiate between the samples of different classifications on key components with ROAV ≥ 1, PCA [29] and OPLS-DA [30] were applied, as shown in Figure 4. According to Figure 4A,C, pomegranate seed samples were clustered in PC1 and PC2, which correspond to 67.9% and 15.6% of the explained related variance of the data, respectively. The distribution of samples on the *X*-axis from right to left with the temperature rise is obvious, implying that the samples were clustered within groups and separated between groups. The principal component score plot with the loading plot was used to intuitively show the correlation between pomegranate seed samples and VOCs. As shown in Figure 4B, the group CK was in the same quadrant as six components including nonanal, octanal, hexanal (monomer), 2-methylbutyl acetate (monomer and dimer), and ethyl acetate, but was more strongly related to that of No.31 (ethyl acetate). Group A was in the same quadrant with the four components of hexanal (monomer), ethyl valerate (monomer and dimer), and 1-propanethiol, particularly close to No.52 (1-propanethiol). Groups B and C, including pentanal, 2-methyl-1-butanal, 3-methyl-1-butanal, butyraldehyde, and (E)-pentene-2-one (dimer), were all located in the same quadrant. In group D, γ-butyrolactone and dimethyl disulfide were two components in the same quadrant. The above results demonstrate that the ester flavor contributed more to the CK and A groups. The aldehydes’ flavor contributed more to CK, A, B, and C, with the heterocyclic compounds contributing more to the flavor of group D. Figure 4D illustrates the verification diagram of the OPLS-DA model, where the Q2 regression line intersects the abscissa and intersects the y-coordinate at an intercept less than 0, with the significance probability value of *p* < 0.05. This result shows that the OPLS-DA model was reliable and statistically significant in this study. The variable importance in projection (VIP) value of the OPLS-DA model was used to screen out the key chemical components and highlight the key aroma components of different groups in pomegranate seeds, as shown in Figure 5. At 95% confidence intervals, there were 7 components with VIP values greater than 1 in pomegranate seeds at different temperatures, including γ-butyrolactone, (E)-3-penten-2-one (dimer), pentanal, 1-propanethiol, octanal, and ethyl valerate (monomer). This result indicated that these seven components were the different components that distinguish pomegranate seeds at different temperatures. This finding is similar to the result of fingerprint spectrum and ROAV method analysis.

## 3. Discussion

Lipid oxidation and Maillard reactions are two of the most common chemical reactions that promote flavor formation. Both reactions follow the production of key intermediates, which are subsequently composed, degraded, recycled, and rearranged to produce volatile flavor compounds [31]. In addition, sugar degradation, the thermal degradation of amino acids, and a lipid oxidation–Maillard reaction also contribute to the formation of flavor. The mechanism of pomegranate seed flavor generation at different processing temperatures is shown in Figure 6.

Lipids play an important role in the production of volatile flavor compounds, and lipid oxidation is one of the reactions involved in the formation of characteristic flavor components. The lipid composition of pomegranate seeds accounted for 20%, which was mainly composed of punicic acid, oleic acid, linoleic acid, linolenic acid, and other unsaturated fatty acids [4]. At high temperatures, lipids are first degraded to fatty acids, then further oxidized to generate hydroperoxides, and eventually converted to volatile carbonyl compounds, such as aldehydes, ketones, esters, alcohols, and furans [15]. In general, stearic acid can be converted to 2-butylfuran, oleic acid to pentanal (-ol), nonanal and octanal, linoleic acid to n-hexanal (-ol), pentanal (-ol), 2-butylfuran and benzaldehyde, and linolenic acid to octanal, nonanal, hexanal, butyraldehyde, and (E)-2-hexenal [31,32,33]. This result is basically consistent with the result of Özcan [9]; that is, the variation in fatty acid content determined by baking. In addition, punicic acid, which accounts for a relatively high proportion of pomegranate seeds, is a scarcely conjugated trienoic acid, and reports of its degradation mechanism and degradation products are rare.

Another important reaction in flavor formation is the Maillard reaction, which is known as an amino-acid-reducing sugar reaction with a cascade of highly complex chemical reactions. In the reaction process, a series of products are produced through condensation, polymerization, degradation, and cyclysis, including ketones, aldehydes, alcohols, furans and their derivatives, and aromatic compounds. Nevertheless, nitrogenous and sulfur-containing compounds are predominant [31,32,34]. The Maillard reaction may also occur during the processing of pomegranate seeds, and the reaction products are similar to amino acid degradation.

Sugar degradation contributes to the development of flavor. The content of crude polysaccharide in pomegranate seeds was about 2.8% [35], which decreased with the increase in the temperature. On the one hand, a caramelization reaction occurs to produce furfural; on the other hand, polysaccharides can also be converted into reducing sugars to further participate in the Maillard reaction. In addition, the thermal degradation of amino acids is one of the pathways of flavor formation. Pomegranate seeds’ protein content is about 20%, which is rich in amino acids [5]. In addition, leucine can be degraded into 2-methyl-1-butanal (-ol) and 3-methyl-1-butanal (-ol), but isoleucine is degraded into 2-methyl-1-butanal (-ol). Valine can be degraded to methacrolein. Benzaldehyde is produced by the degradation of phenylalanine. Additionally, glutamic acid can be degraded into acetone, γ-butyrolactone, 2-butanone, furfural, pyrrole, etc. Pyrrole may also be produced by the degradation of arginine and histidine; 2,5-dimethylpyrazine can be produced by serine degradation [31,36].

The lipid–Maillard reaction is another means of flavor formation, which mainly generates nitrogen-containing, sulfur-containing, and other heterocyclic compounds [33,37]. The interactions between the lipid and Maillard reaction products are the sources of many flavor compounds; 2,5-dimethylpyrazine, trimethylpyrazine, pyrrole, dimethyl disulfide, and 1-propanethiol can be produced by this pathway.

## 4. Materials and Methods

### 4.1. Sample Preparation

Pomegranate samples were gathered from the Yicheng pomegranate Garden (N: 34°46′45.53″, E: 117°32′32.63″) in October 2021. Sour pomegranate fruit of the Daqingpi variety was peeled and juiced by hand, and the pomegranate seeds were removed and then scrubbed repeatedly until there was no pulp residue. After that, the seeds were dried at 50 °C in an electro-thermostatic blast oven (Jinghong Experimental Equipment Co., Ltd., Shanghai, China) and stored at −20 °C in a low-temperature refrigerator (Qingdao Haier Special Refrigerator Co., Ltd., Qingdao, China). The 125 g sample was randomly divided into 5 groups, including the CK group without any treatment, and the A, B, C, and D groups, which were processed at 140, 160, 180, and 200 °C, respectively, for 20 min in a temperature-controlled electric wok (Xinyang Yiding Tea Technology Co., Ltd., Xinyang, China) [38]. All samples were stored at −20 °C in frozen storage to prepare for the subsequent analyses.

### 4.2. GC-IMS Analysis

Measurements were made using a GC-IMS instrument (FlavourSpec^®^) from the G.A.S. Department of Shandong HaiNeng Science Instrument Co., Ltd. (Shandong, China). The GC was equipped with an MXT-5 capillary column (15 m × 0.53 mm ID, 1 µm). The GC column temperature was 60 °C, and the IMS temperature was 45 °C. Then, 3.00 g of finely ground samples was transferred into a 20 mL headspace glass sampling vial and incubated at 60 °C for 15 min. Moreover, 500 µL of headspace samples was automatically injected into the injector (85 °C, splitless mode) by means of a heated syringe. Nitrogen of 99.999% purity was used as the carrier/drift gas at programmed flow rates as follows: EPC1 (IMS drift gas) was maintained at 150 mL·min^−1^, EPC2 (GC carrier gas) was initially maintained at 2 mL·min^−1^ for 2 min and then increased to 100 mL·min^−1^ within 18 min [39,40]. N-ketones C_4_-C_9_ (Sinopharm Chemical Reagent Co., Ltd., Beijing, China) were used as external references to calculate the retention index (RI) of volatile compounds. Volatile compounds were identified by comparing RI and the drift time (DT, which is the time it takes for ions to reach the collector through drift tube, in milliseconds) of the standard in the GC-IMS library.

### 4.3. Relative Odor Activity (ROAV)

The relative odor activity value (ROAV_i_) is used to evaluate the contribution of individual compounds to the overall aroma [12]. It was calculated as follows:$$\begin{array}{l} {\mathrm{ROVA}}_{i}=\frac{C_{i}}{T_{i}}\times \frac{T_{\max}}{C_{\max}}\times 100 \end{array}$$
where C_i_ is the relative content of the volatile compound to be measured (%), and T_i_ is the aroma threshold of the volatile compounds as available in the literature. Additionally, T_max_ and C_max_ are the maximum of C_i_/T_i_ among all the compounds in the sample. Volatile compounds with ROAV ≥ 1 are considered key odor compounds, of which 1 > ROAV > 0.1 plays an important role in aroma; those <0.1 are potential aroma compounds [12].

### 4.4. Statistical Analysis

All samples were analyzed in triplicate. The GC-IMS Library Search can perform the qualitative identification of detected substances according to the built-in NIST database and IMS database. Reporter and Gallery Plot plug-ins were used to analyze the differences between sample spectra and to compare fingerprint patterns. SIMCA-P14.1 software was used to perform principal component analysis (PCA) and orthogonal partial least squares discrimination analysis (OPLS-DA). Additionally, the one-way analysis of variance (ANOVA) was used to compare the significance between groups (SPSS 26.0), *p* < 0.05.

## 5. Conclusions

A total of 54 types of VOCs were identified by GC-IMS technology in pomegranate seed samples. It was found that different processing temperatures lead to different VOCs. In total, 11, 13, 15, 17, and 15 key flavor components of groups CK, A, B, C, and D, respectively, were analyzed by the ROAV method. PCA and OPLS-DA analysis were used to identify seven differentially contributing components in pomegranate seed samples, including γ-butyrolactone, (E)-3-penten-2-one (dimer), pentanal, 1-propanethiol, octanal, and ethyl valerate (monomer). The results showed that lipid oxidation and the Maillard reaction may be the main mechanisms of flavor formation during the processing of pomegranate seeds. This is of great significance for the study of pomegranate seed flavor and also lays a theoretical foundation for further the research or development of pomegranate seed flavor products.

## Figures and Tables

**Figure 1 molecules-28-02717-f001:**
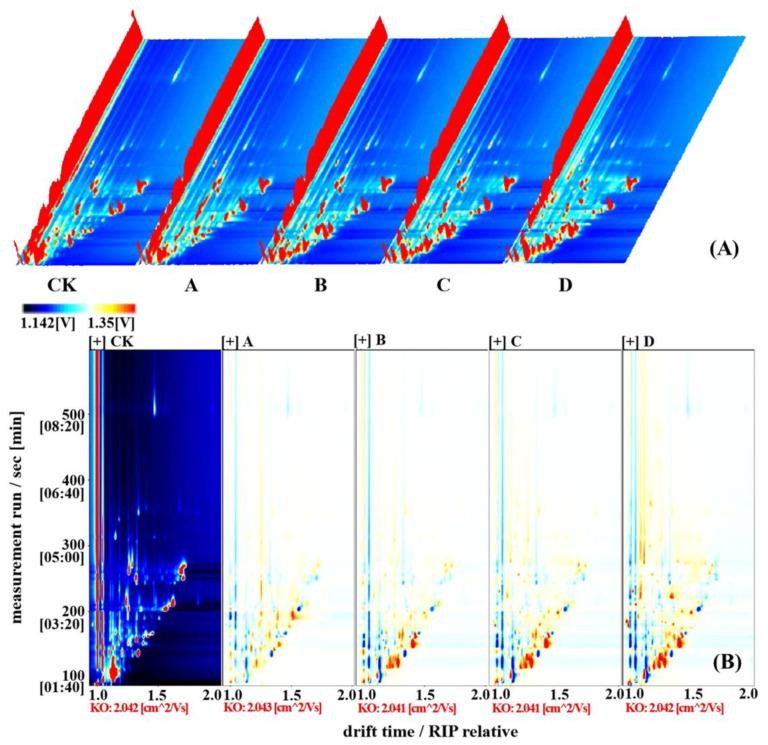
Topographic plot of pomegranate seeds at different processing temperatures. (**A**) 3D topographic plots of VOCs in pomegranate seeds at different processing temperatures; (**B**) 2D topographic plots of VOCs in pomegranate seeds at different processing temperatures (note: CK, A, B, C, and D represent raw pomegranate seeds, and pomegranate seeds processed at 140, 160, 180, and 200 °C, respectively. The same is the case below.).

**Figure 2 molecules-28-02717-f002:**
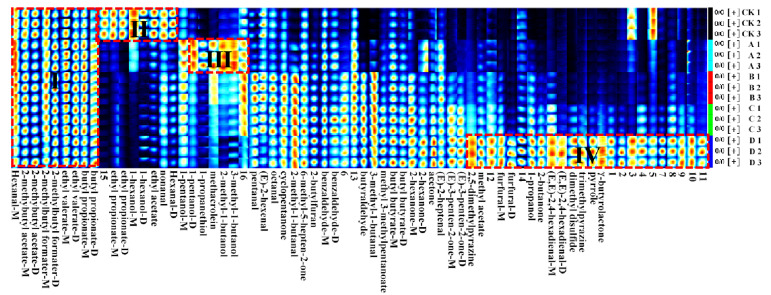
Fingerprints of volatile compounds in pomegranate seeds at different processing temperatures.

**Figure 3 molecules-28-02717-f003:**
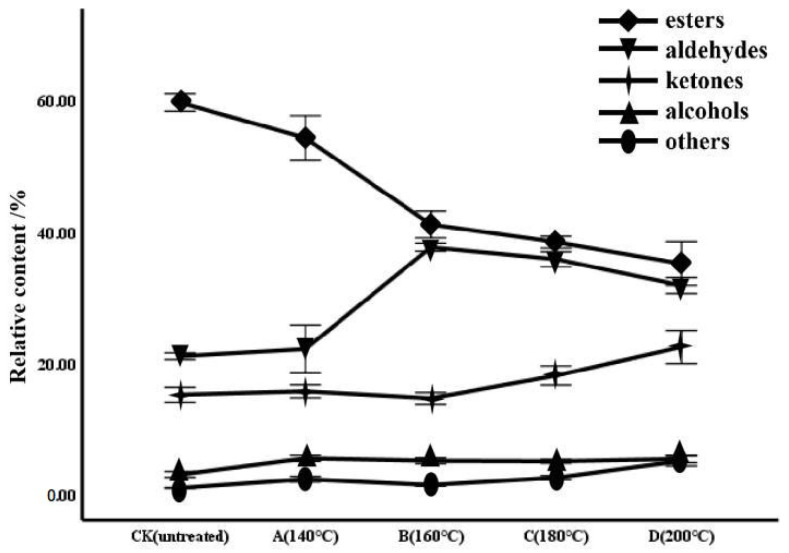
Trend diagram of the relative content change of various components in pomegranate seeds at different processing temperatures.

**Figure 4 molecules-28-02717-f004:**
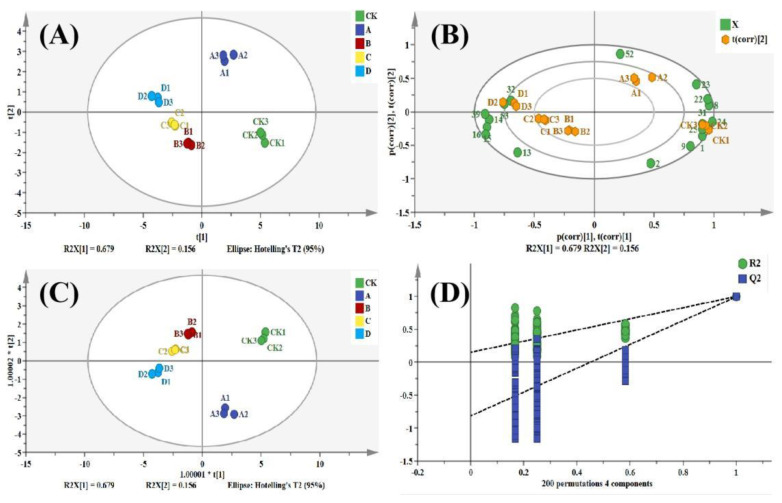
PCA and OPLS−DA analysis of key flavor components in pomegranate seeds at different processing temperatures. (**A**) PCA score scatter plot; (**B**) PCA biplot; (**C**) OPLS-DA score scatter plot; (**D**) OPLS-DA model validation plot.

**Figure 5 molecules-28-02717-f005:**
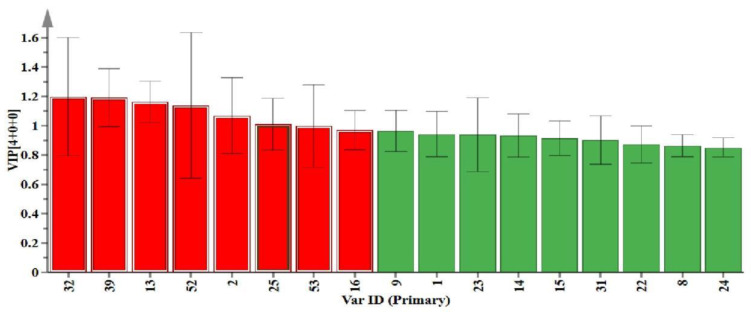
The variable importance in projection (VIP) value of the OPLS-DA model.

**Figure 6 molecules-28-02717-f006:**
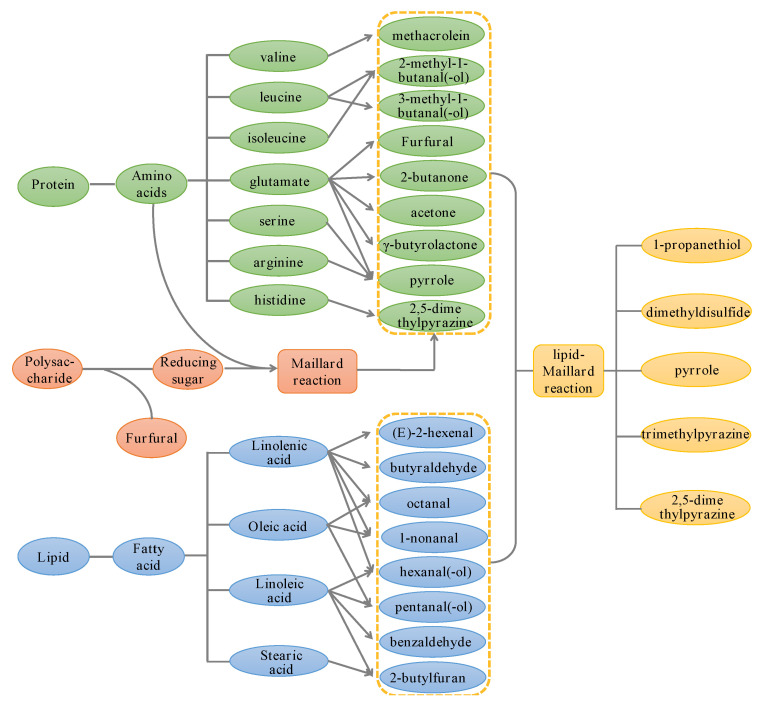
Mechanism of flavor formation in pomegranate seed processing.

**Table 1 molecules-28-02717-t001:** HS-GC-IMS integration parameters of pomegranate seeds under different Paozhi processing conditions.

No.	Temperatures. Compound	CAS #	RI^1^	RT^2^	DT^3^	Relative Amount %				
						CK	A	B	C	D
1	nonanal	C124196	1108.7	507.230	1.4748	1.7575 ± 0.0437 ^a^	0.9254 ± 0.0415 ^b^	0.8768 ± 0.0721 ^b^	0.8310 ± 0.0648 ^b^	0.5906 ± 0.0449 ^c^
2	octanal	C124130	1004.8	357.909	1.4042	0.4450 ± 0.0419 ^a^	0.3243 ± 0.0160 ^c^	0.4492 ± 0.0113 ^a^	0.3826 ± 0.0026 ^b^	0.2788 ± 0.0174 ^d^
3	benzaldehyde	C100527	959.6	314.471	1.1510	1.7815 ± 0.0265 ^a^	1.8515 ± 0.1158 ^a^	1.3131 ± 0.1053 ^b^	1.4545 ± 0.0456 ^b^	1.0814 ± 0.0763 ^d^
4	benzaldehyde *	C100527	957	312.223	1.4707	0.2333 ± 0.0168 ^a^	0.2879 ± 0.0420 ^a^	0.1719 ± 0.0288 ^b^	0.2349 ± 0.0276 ^a^	0.1639 ± 0.0300 ^b^
5	(E)-2-heptenal	C18829555	954.1	309.782	1.2585	0.4368 ± 0.0158 ^c^	0.9670 ± 0.0222 ^a^	0.7591 ± 0.0575 ^b^	0.6748 ± 0.0335 ^b^	0.6823 ± 0.0710 ^b^
6	(E,E)-2,4-hexadienal	C142836	913.4	275.048	1.1134	0.0831 ± 0.0059 ^d^	0.0972 ± 0.0097 ^d^	0.1631 ± 0.0120 ^c^	0.3616 ± 0.0100 ^a^	0.3365 ± 0.0155 ^b^
7	(E,E)-2,4-hexadienal *	C142836	912.8	274.508	1.4565	0.0268 ± 0.0011 ^b^	0.0189 ± 0.0005 ^b,c^	0.0171 ± 0.003 ^b,c^	0.0206 ± 0.0039 ^c^	0.0833 ± 0.0086 ^a^
8	hexanal	C66251	794.4	204.500	1.2590	2.2416 ± 0.0914 ^a^	2.1485 ± 0.1628 ^a^	1.6343 ± 0.0264 ^b^	1.3241 ± 0.0538 ^c^	0.9844 ± 0.0222 ^d^
9	hexanal *	C66251	794.4	204.500	1.5656	7.9134 ± 0.2000 ^a^	1.8638 ± 0.0897 ^d^	2.8157 ± 0.0712 ^b^	2.2486 ± 0.0945 ^c^	1.3543 ± 0.0957 ^e^
10	(E)-2-hexenal	C6728263	846.1	232.218	1.1849	0.6842 ± 0.0074 ^a^	0.4589 ± 0.0111 ^c^	0.5444 ± 0.0260 ^b^	0.5170 ± 0.0278 ^b^	0.2520 ± 0.0208 ^d^
11	furfural	C98011	830.4	223.806	1.0834	0.1538 ± 0.0331 ^d^	0.2401 ± 0.0451 ^d^	0.5524 ± 0.0041 ^c^	0.8186 ± 0.1061 ^b^	1.5428 ± 0.0647 ^a^
12	furfural *	C98011	828.8	222.948	1.34332	0.5856 ± 0.0168 ^b,c^	0.5671 ± 0.0540 ^b,c^	0.4350 ± 0.0178 ^c^	0.6909 ± 0.1040 ^b^	2.7142 ± 0.2395 ^a^
13	pentanal	C110623	695.7	163.151	1.4254	2.6950 ± 0.1009 ^d^	2.4390 ± 0.1282 ^e^	5.5583 ± 0.0201 ^a^	4.3723 ± 0.1165 ^b^	3.8085 ± 0.0729 ^c^
14	2-methyl-1-butanal	C96173	665.9	154.182	1.4021	1.0585 ± 0.0407 ^e^	3.9203 ± 0.6797 ^d^	7.1892 ± 0.0975 ^a^	6.5440 ± 0.1397 ^b^	5.7841 ± 0.0439 ^c^
15	3-methyl-1-butanal	C590863	641.5	147.599	1.4079	0.2557 ± 0.0120 ^e^	2.0161 ± 0.4195 ^d^	5.3988 ± 0.0305 ^a^	4.9995 ± 0.1126 ^b^	4.2255 ± 0.0691 ^c^
16	butyraldehyde	C123728	550.4	123.005	1.2835	0.1776 ± 0.0230 ^e^	1.4384 ± 0.5071 ^d^	8.3895 ± 0.2514 ^b^	9.5444 ± 0.1290 ^a^	7.4709 ± 0.1027 ^c^
17	methacrolein	C78853	551.4	123.299	1.2169	0.5279 ± 0.0327 ^d^	2.5842 ± 0.2809 ^a^	1.3495 ± 0.0321 ^b^	0.7818 ± 0.0039 ^c^	0.4465 ± 0.0141 ^d^
	aldehydes	NA	NA	NA	NA	21.0570 ± 0.2003 ^d^	22.1486 ± 1.4635 ^d^	37.6173 ± 0.2398 ^a^	35.8013 ± 0.4550 ^b^	31.800 ± 0.4862 ^c^
18	butyl butyrate	C109217	1000.3	351.460	1.3425	1.3016 ± 0.0238 ^c^	1.6518 ± 0.0554 ^a^	1.3822 ± 0.1288 ^b,c^	1.5011 ± 0.0357 ^b^	1.3413 ± 0.1402 ^b,c^
19	butyl butyrate *	C109217	1000	350.939	1.8248	0.1477 ± 0.0187 ^b^	0.2256 ± 0.0244 ^a^	0.2084 ± 0.0394 ^a,b^	0.2641 ± 0.0149 ^a^	0.2640 ± 0.0587 ^a^
20	butyl propionate	C590012	910.2	272.272	1.2888	2.7695 ± 0.0538 ^b^	3.2107 ± 0.1691 ^a^	2.5648 ± 0.0422 ^c^	2.1454 ± 0.0722 ^d^	1.7344 ± 0.0991 ^e^
21	butyl propionate *	C590012	906.5	269.147	1.7242	3.1779 ± 0.1337 ^b^	4.3391 ± 0.0827 ^a^	3.1282 ± 0.1828 ^b^	2.9590 ± 0.0450 ^b^	2.5499 ± 0.3002 ^c^
22	ethyl valerate	C539822	896.1	260.254	1.2659	4.9549 ± 0.1298 ^a^	4.9741 ± 0.2239 ^a^	3.7620 ± 0.0635 ^b^	3.3694 ± 0.0734 ^c^	2.7120 ± 0.0744 ^d^
23	ethyl valerate *	C539822	896.6	260.683	1.6882	9.7959 ± 0.1747 ^b^	11.1123 ± 0.4489 ^a^	7.8986 ± 0.2613 ^c^	7.512 ± 0.0045 ^c^	6.2828 ± 0.3407 ^d^
24	2-methybutyl acetate	C624419	880	250.392	1.3196	3.8453 ± 0.1007 ^a^	2.8518 ± 0.1506 ^b^	2.4201 ± 0.0711 ^c^	1.9358 ± 0.0670 ^d^	1.5570 ± 0.1008 ^e^
25	2-methybutyl acetate *	C624419	880.4	250.606	1.7082	4.0193 ± 0.1333 ^a^	2.4034 ± 0.0471 ^b^	2.2326 ± 0.1351 ^b^	1.7157 ± 0.0561 ^c^	1.8974 ± 0.2845 ^c^
26	methyl 3-methylpentanoate	C2177788	870.8	245.460	1.2271	0.2404 ± 0.0192 ^c^	0.3400 ± 0.0107 ^b^	0.3387 ± 0.0238 ^b^	0.4909 ± 0.0244 ^a^	0.5142 ± 0.0220 ^a^
27	2-methylbutyl formater	C35073279	813.3	214.651	1.2386	3.1476 ± 0.0838 ^a^	2.6039 ± 0.1117 ^b^	2.1905 ± 0.0451 ^c^	1.8812 ± 0.0215 ^d^	1.5340 ± 0.0131 ^e^
28	2-methylbutyl formate *	C35073279	810.9	213.382	1.6259	12.816 ± 0.1743 ^b^	14.9349 ± 0.3815 ^a^	11.1614 ± 0.1967 ^c^	10.4572 ± 0.2321 ^d^	8.5892 ± 0.5250 ^e^
29	ethyl propionate	C105373	709.5	168.783	1.1543	1.0471 ± 0.0190 ^a^	0.2206 ± 0.0399 ^b^	0.1659 ± 0.0049 ^c^	0.2099 ± 0.0129 ^b^	0.2223 ± 0.0111 ^b^
30	ethyl propionate *	C105373	707.5	167.948	1.4550	1.7082 ± 0.0391 ^a^	0.0964 ± 0.0115 ^c^	0.1784 ± 0.0174 ^b^	0.1351 ± 0.0095 ^c^	0.1182 ± 0.0049 ^c^
31	ethyl acetate	C141786	603.8	137.409	1.3388	9.8285 ± 0.1269 ^a^	3.5882 ± 0.0799 ^b^	1.5800 ± 0.0718 ^c^	1.3840 ± 0.0598 ^c^	1.4688 ± 0.2830 ^c^
32	γ-butyrolactone	C96480	925	284.914	1.0873	0.2802 ± 0.0197 ^c^	0.2850 ± 0.0134 ^c^	0.3137 ± 0.0244 ^c^	0.5313 ± 0.0535 ^b^	1.4067 ± 0.2259 ^c^
33	methyl acetate	C79209	533.3	118.410	1.1950	0.5375 ± 0.0572 ^d^	1.3797 ± 0.1958 ^c^	1.5426 ± 0.0695 ^c^	1.8951 ± 0.0701 ^b^	2.9349 ± 0.1170 ^a^
	esters	NA	NA	NA	NA	59.6174 ± 0.5287 ^a^	54.2175 ± 1.3696 ^b^	41.0682 ± 0.8210 ^c^	38.3872 ± 0.3593 ^d^	35.1271 ± 1.3307 ^e^
34	6-methyl-5-hepten-2-one	C110930	991.4	341.561	1.1813	0.5666 ± 0.0316 ^a^	0.4972 ± 0.0382 ^b^	0.4504 ± 0.0214 ^b,c^	0.4074 ± 0.0195 ^c,d^	0.3741 ± 0.0239 ^d^
35	2-hexanone	C591786	784.1	199.001	1.1906	0.2423 ± 0.0081 ^d^	0.1453 ± 0.0155 ^e^	0.4042 ± 0.0075 ^c^	0.4773 ± 0.0095 ^a^	0.4298 ± 0.0045 ^b^
36	2-hexanone *	C591786	784.9	199.424	1.5043	0.2213 ± 0.0012 ^d^	1.4360 ± 0.0959 ^a^	0.5284 ± 0.0160 ^c^	0.8977 ± 0.0658 ^b^	1.0059 ± 0.0634 ^b^
37	cyclopentanone	C120923	784.9	199.424	1.1088	0.2652 ± 0.0226 ^c^	0.2693 ± 0.0131 ^c^	0.3637 ± 0.0108 ^a^	0.2965 ± 0.0040 ^b^	0.2680 ± 0.0092 ^c^
38	(E)-3-penten-2-one	C3102338	740.9	181.506	1.0950	0.0636 ± 0.0086 ^d^	0.0437 ± 0.0055 ^e^	0.1155 ± 0.0021 ^c^	0.2096 ± 0.0115 ^b^	0.2342 ± 0.0094 ^a^
39	(E)-3-penten-2-one *	C3102338	739.4	180.880	1.3544	0.0524 ± 0.0032 ^d^	0.0954 ± 0.0003 ^d^	0.2152 ± 0.0225 ^c^	0.5012 ± 0.0344 ^b^	0.5672 ± 0.0375 ^a^
40	2-butanone	C78933	579.1	130.746	1.2493	0.2867 ± 0.0356 ^d^	0.4217 ± 0.0512 ^d^	1.4987 ± 0.0625 ^c^	2.8423 ± 0.1902 ^b^	4.8303 ± 0.3168 ^a^
41	acetone	C67641	488.2	106.250	1.1168	13.5036 ± 0.4365 ^b^	12.8166 ± 0.3529 ^b,c^	11.0784 ± 0.2844 ^d^	12.4838 ± 0.2770 ^c^	14.7516 ± 0.6146 ^a^
	ketones	NA	NA	NA	NA	15.2017 ± 0.4559 ^c^	15.7252 ± 0.3970 ^c^	14.6545 ± 0.3551 ^c^	18.1158 ± 0.5860 ^b^	22.4611 ± 1.0116 ^a^
42	1-hexanol	C111273	867.6	243.760	1.3264	0.2803 ± 0.0500 ^a^	0.1518 ± 0.0292 ^b^	0.0410 ± 0.0005 ^c^	0.0245 ± 0.0039 ^b^	0.0178 ± 0.0025 ^c^
43	1-hexanol *	C111273	867	243.423	1.6423	0.5629 ± 0.0061 ^a^	0.1163 ± 0.0138 ^b^	0.0520 ± 0.0040 ^c^	0.0774 ± 0.0076 ^c^	0.1124 ± 0.0087 ^b^
44	1-pentanol	C71410	767.7	192.352	1.2549	1.1628 ± 0.0739 ^c^	1.8889 ± 0.0468 ^a^	1.3912 ± 0.0169 ^b^	0.9957 ± 0.0371 ^d^	0.7058 ± 0.0240 ^e^
45	1-pentanol *	C71410	765.1	191.310	1.5111	0.6755 ± 0.0825 ^c^	1.5449 ± 0.0697 ^a^	0.8298 ± 0.0148 ^b^	0.6302 ± 0.0349 ^c^	0.4774 ± 0.0225 ^d^
46	2-methyl-1-butanol	C137326	736.1	179.543	1.2325	0.0654 ± 0.0087 ^b^	0.1621 ± 0.0370 ^a^	0.0740 ± 0.0081 ^b^	0.0402 ± 0.0025 ^b^	0.0426 ± 0.0019 ^b^
47	3-methyl-1-butanol	C123513	733.4	178.437	1.2455	0.1050 ± 0.0166 ^c^	0.3292 ± 0.0151 ^a^	0.1361 ± 0.0057 ^b^	0.1167 ± 0.0066 ^b,c^	0.1028 ± 0.0064 ^c^
48	1-propanol	C71238	533.3	118.400	1.2406	0.2436 ± 0.0177 ^e^	1.3508 ± 0.2507 ^d^	2.6556 ± 0.1400 ^c^	3.2128 ± 0.1190 ^b^	3.992 ± 0.1698 ^a^
	alcohols	NA	NA	NA	NA	3.0955 ± 0.1819 ^d^	5.5440 ± 0.1867 ^a^	5.1797 ± 0.1730 ^b,c^	5.0974 ± 0.1131 ^c^	5.4508 ± 0.2240 ^a,b^
49	trimethylpyrazine	C14667551	1009.4	364.484	1.1689	0.2055 ± 0.0165 ^c^	0.2092 ± 0.0144 ^c^	0.1904 ± 0.0323 ^b^	0.3377 ± 0.0252 ^b^	0.7746 ± 0.0760 ^a^
50	2,5-dimethylpyrazine	C123320	925.7	285.492	1.1143	0.1579 ± 0.0202 ^d^	0.2005 ± 0.0104 ^c^	0.2123 ± 0.0074 ^c^	0.2786 ± 0.0108 ^b^	0.3419 ± 0.0141 ^a^
51	pyrrole	C109977	750.7	185.469	0.9712	0.1629 ± 0.0037 ^c^	0.1643 ± 0.0058 ^c^	0.1508 ± 0.0021 ^c^	0.3876 ± 0.0245 ^b^	1.3089 ± 0.0817 ^a^
52	1-propanethiol	C107039	622.3	142.406	1.3625	0.1521 ± 0.0169 ^b^	1.3462 ± 0.1984 ^a^	0.3207 ± 0.0260 ^b^	0.2908 ± 0.0059 ^b^	0.2785 ± 0.0200 ^b^
53	dimethyl disulfide	C624920	735.3	179.212	0.9860	0.1857 ± 0.0166 ^c^	0.1459 ± 0.0129 ^c^	0.2181 ± 0.0045 ^c^	0.8543 ± 0.0682 ^b^	2.0297 ± 0.1241 ^a^
54	2-butylfuran	C4466244	890	255.752	1.1821	0.1639 ± 0.0043 ^d^	0.2986 ± 0.0231 ^c^	0.3880 ± 0.0232 ^b^	0.4494 ± 0.0253 ^a^	0.4274 ± 0.0235 ^a^
	others	NA	NA	NA	NA	1.0279 ± 0.0134 ^d^	2.3647 ± 0.1720 ^b^	1.4803 ± 0.0743 ^c^	2.5983 ± 0.1262 ^b^	5.1610 ± 0.3071 ^a^

# represents the chemical substance registration number; * Dimers representing compounds; RI^1^ represents the residence time in the capillary gas chromatography column; RT^2^ represents the retention index calculated with n-ketone C_4_–C_9_ as the external standard on fs-se-54-cb column; DT^3^ represents the drift time of the drift tube; NA indicates not detected; ^a–e^ mean values are significantly different in rows (*p* < 0.05).

## Data Availability

Data are contained within the article.

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
