# Peer review of "Analysis of the Volatile Flavor Compounds of Pomegranate Seeds at Different Processing Temperatures by GC-IMS"

_molecules, 2023, doi:10.3390/molecules28062717_

Round 1

Reviewer 1 Report

comments on the manuscript

Reviewer 2 Report

This study has concentrated on Analysis of Volatile Flavor Compounds of Pomegranate Seeds 2

at Different Processing Temperature by GC-IMS.

The authors should work on the following questions-

§  The introduction part of the manuscript is very weak and requires drastic improvement. There are many grammatical mistakes in this part.

§  The introduction part should compile critical findings from the preceding studies and highlight the research gaps.

§  Add GC-IMS spectrum.

§  Figure 6, if copy from the paper. At least the paper should be cited as adopted from (reference) or copy right permission should be granted.

§  The reference style should be revised and be in accordance with the journal format. The years in some references are bold while not in others.

Author Response

  • The introduction part of the manuscript is very weak and requires drastic improvement. There are many grammatical mistakes in this part.

Response: For the introduction part, we have reorganized and examined the grammar.

  • The introduction part should compile critical findings from the preceding studies and highlight the research gaps.

Response: We have further summarized the research status of pomegranate seeds and highlighted gaps with our study.

  • Add GC-IMS spectrum.

Response: We think the GC-IMS spectrum in the article can illustrate our problem.

  • Figure 6, if copy from the paper. At least the paper should be cited as adopted from (reference) or copy right permission should be granted.

Response: Figure 6 is our inference graph based on the relevant reference, which is original and does not involve infringement.

  • The reference style should be revised and be in accordance with the journal format. The years in some references are bold while not in others.

Response: The references were derived from EndNote. We find that molecular journals cite journals with bold years, whereas books, proceedings, and others are not bolded. So the years are bold in some references while not in others.

Reviewer 3 Report

·      Introduction

Please mention in the introduction the importance of pomegranate seed as an interesting source of rare Conjugated Linoleic Fatty Acids, that have interesting pharmacological applications.

Please indicate in the introduction the idea behind the temperature treatment of pomegranate seeds.

·      R&D

All abbreviations used (CK, A, B, C, D, RI, DT, RT, ...) in tables and figures should be defined in the table note or figure caption.

Please remove NA (not detected) in table1 and 2 for family compounds (aldehydes, esters, alcohols, …).

Please check the line 230.

·      M&M

Please add GPS data of plant material location harvest.

How authors did choose the temperature treatment conditions; 140℃, 160℃, 180℃, and 200℃ for 20 min.

How authors did choose the Headspace extraction, and GC separation conditions.

Relative quantification should be done using a detector which consider response factor.

Author Response

Introduction

Please mention in the introduction the importance of pomegranate seed as an interesting source of rare Conjugated Linoleic Fatty Acids, that have interesting pharmacological applications.

 Response: We have mentioned in the introduction the importance of pomegranate seed as an interesting source of rare Conjugated Linoleic Fatty Acids, that have interesting pharmacological applications.

Please indicate in the introduction the idea behind the temperature treatment of pomegranate seeds.

 Response: We have indicated in the introduction the idea behind the temperature treatment of pomegranate seeds.

  • R&D

All abbreviations used (CK, A, B, C, D, RI, DT, RT, ...) in tables and figures should be defined in the table note or figure caption.

Response: We have defined the abbreviations which we used (CK, A, B, C, D, RI, DT, RT in the table note or figure caption.

Please remove NA (not detected) in table1 and 2 for family compounds (aldehydes, esters, alcohols, …).

Response: As the compounds also correspond to other data information, NA in Tables 1 and 2 cannot be removed.

Please check the line 230.

Response: Because of our mistake, line 230 is not complete, and we have made the necessary changes.

  • M&M

Please add GPS data of plant material location harvest.

Response:  We have added GPS data at the relevant locations of the articles in this paper.

How authors did choose the temperature treatment conditions; 140℃, 160℃, 180℃, and 200℃ for 20 min.

Response: In the course of our experiments, we found that pomegranate seeds produce an aromatic component when roasted. According to related literature, temperature can affect odor changes. Therefore, in order to further explore the production mechanism of aroma components in pomegranate seeds, temperature conditions with different gradients were set in the pre-experiment. However, groups with less significant odor changes are not included in this paper.

How authors did choose the Headspace extraction, and GC separation conditions.

 Response: Because of our mistake, the cited references are omitted from the article. We have cited the relevant references to illustrate the choice of headspace extraction and GC separation conditions.

Relative quantification should be done using a detector which consider response factor.

Response: The analysis software VOCal for the GC-IMS instrument can be used for relative quantitative analysis. The relevant literature is reported as follows: Guo, S.; Zhao, X.; Ma, Y.; Wang, Y.; Wang, D. Fingerprints and changes analysis of volatile compounds in fresh-cut yam during yellowing process by using HS-GC-IMS. Food Chem. 2022, 369, 130939, https://doi.org/10.1016/j.foodchem.2021.130939.